# Effectiveness of Immunotherapy in Non-Small Cell Lung Cancer Patients with a Diagnosis of COPD: Is This a Hidden Prognosticator for Survival and a Risk Factor for Immune-Related Adverse Events?

**DOI:** 10.3390/cancers16071251

**Published:** 2024-03-22

**Authors:** Silvia Riondino, Roberto Rosenfeld, Vincenzo Formica, Cristina Morelli, Giusy Parisi, Francesco Torino, Sabrina Mariotti, Mario Roselli

**Affiliations:** Medical Oncology Unit, Department of Systems Medicine, Tor Vergata University, 00133 Rome, Italy; roberto.rosenfeld88@gmail.com (R.R.); vincenzo.formica@uniroma2.it (V.F.); cristina.morelli@ptvonline.it (C.M.); giusy.parisi@ptvonline.it (G.P.); torino@med.uniroma2.it (F.T.); sabrina.mariotti@ptvonline.it (S.M.); mario.roselli@uniroma2.it (M.R.)

**Keywords:** non-small cell lung cancer, chronic obstructive pulmonary disease, immunotherapy

## Abstract

**Simple Summary:**

The interaction between the immune system, chronic obstructive pulmonary disease (COPD), and non-small cell lung cancer (NSCLC) is complex and multifaceted and involves all cellular elements of the tumour microenvironment, together with the molecules expressed and secreted in the inflamed milieu. In patients with both diseases, considering that COPD is thought to impair the immune response against tumour cells, immune checkpoint inhibitors (ICIs) combined with chemotherapy appear to improve the pathological responses of NSCLC patients, showing promising improvements in survival. In the present review, we sought to understand the interaction between the two pathways and how the efficacy of immunotherapy in patients with NSCLC and COPD is affected in these patients.

**Abstract:**

The interplay between the immune system and chronic obstructive pulmonary disease (COPD) and non-small cell lung cancer (NSCLC) is complex and multifaceted. In COPD, chronic inflammation and oxidative stress can lead to immune dysfunction that can exacerbate lung damage, further worsening the respiratory symptoms. In NSCLC, immune cells can recognise and attack the cancer cells, which, however, can evade or suppress the immune response by various mechanisms, such as expressing immune checkpoint proteins or secreting immunosuppressive cytokines, thus creating an immunosuppressive tumour microenvironment that promotes cancer progression and metastasis. The interaction between COPD and NSCLC further complicates the immune response. In patients with both diseases, COPD can impair the immune response against cancer cells by reducing or suppressing the activity of immune cells, or altering their cytokine profile. Moreover, anti-cancer treatments can also affect the immune system and worsen COPD symptoms by causing lung inflammation and fibrosis. Immunotherapy itself can also cause immune-related adverse events that could worsen the respiratory symptoms in patients with COPD-compromised lungs. In the present review, we tried to understand the interplay between the two pathologies and how the efficacy of immunotherapy in NSCLC patients with COPD is affected in these patients.

## 1. Introduction

Chronic obstructive pulmonary disease (COPD) and non-small cell lung cancer (NSCLC) are two chronic respiratory conditions with significant global impact. Non-small lung cancer (NSCLC) is the second highest tumour in incidence and mortality worldwide [1], remarking it as one of the renowned “big killers”. Recently, it has been estimated that 6.2% (about 1 person out of 16) of the American population has developed this disease in the 3-year interval from 2017 to 2019 and it has been established that 238,340 new cases and 127,070 new deaths will be detected in 2023 [1]. In Europe, there were almost 480,000 new cases of lung cancer (11.8% of all new diagnoses) with more than 380,000 deaths in 2020 [2]. Chronic obstructive pulmonary disease (COPD) is an inflammatory disease affecting alveoli, airways, and microvasculature, characterised by persistent respiratory symptoms and progressive airflow obstruction [3]. COPD can coexist with other lung diseases and precede NSCLC. Besides the known genetic and epigenetic mechanisms of NSCLC development in individuals with COPD [4], COPD and NSCLC, despite their distinct clinical manifestations, share some common molecular pathways, exhibiting overlapping molecular alterations that may contribute to their coexistence or exacerbate their symptoms. Given the inflammatory nature of COPD, in patients where lung cancer developed after a pre-existing obstructive pulmonary condition, immunotherapy might be more effective, possibly because chronic inflammation contributes to increased tumour-infiltrating lymphocytes (TILs). Indeed, the expression of pathways activating T-cell-mediated immune responses through immune checkpoint proteins PD-1 and PD-L1 appears dysregulated in COPD patients [5,6,7]. While the incidence of NSCLC is dropping annually both in males (2.6%) and females (1.1%) [1], COPD is a nosocomial entity with an increase in both sexes [8] often triggered by the same risk factors and, as stated, coexisting with NSCLC in the same patient. The mortality for NSCLC has decreased substantially from the 1990s with a 4% cumulative reduction per year, thanks to the significant advances in treatment options and the wider availability of selectable target therapies toward genetic alterations and immunotherapy. Indeed, the immune system plays a critical role in both COPD and NSCLC, with complex interactions. COPD can impair the immune response against cancer cells, leading to an increased risk of tumour growth and metastasis. On the other hand, NSCLC can evade immune surveillance and create an immunosuppressive environment that worsens the lung inflammation. The discrimination between “self” and “non-self” operated by the immune system is regulated by a delicate balance between immunoregulatory and effector cells which, when altered, elicit immune responses against “self” antigens. Immune checkpoint molecules are involved in preventing reactions against the “self”, thus participating in immune tolerance.

In the scenery of NSCLC immunotherapy, immune checkpoint inhibitors (ICIs), such as Pembrolizumab, Atezolizumab, Cemipilimab, Nivolumab or Ipilimumab, are characterised by important overall response rates of 32–63.5% and 2-year OS rates of 38–45% [9,10,11,12,13]. However, about 20% of patients do not respond to upfront immunotherapy due to unknown factors interfering with the immune response. Thus, the need to identify novel biomarkers that could distinguish between responders and non-responders in order to deliver better treatments to these patients is of great importance. Unfortunately, despite many efforts in this field, they have not yet been elucidated. Moreover, several studies have observed that the presence of systemic chronic inflammation (lupus, rheumatoid arthritis or multiple sclerosis) before immunotherapy starts can lead to an increased risk of immune-related adverse events (irAEs) involving chronically inflamed tissues [14].

A promising potential predictive factor, yet unexplored in clinical trials, is the presence of COPD and its immunological biomarkers. In this narrative review, we present the state of the art on this ever-growing subgroup of patients.

## 2. PD-L1/PD-1 Interaction and CTLA-4 Stimulation: Pathways of Immune Anergy

Immunomodulation is based on two opposite types of signals with a positive and a negative regulation led by costimulatory and co-inhibitory molecules [15]. The same ligand could be stimulatory or inhibitory depending on which receptor is elicited, usually by means of different receptor densities expressed on the membrane surface [15,16], as shown in Figure 1.

The balance between pro-inflammatory and immunosuppressive T-cell responses in lung cancer is critical for disease outcomes. Immunotherapeutic approaches aim to tip this balance in favour of anti-tumour immunity by targeting Treg activation or inhibitory pathways such as the programmed cell death protein 1 (PD-1), its ligands (PD-L1 and PD-L2) [16,17,18] or CTLA-4. PD-1 and CTLA-4 are two different molecules with two different roles in T-cell anergy. Indeed, PD-1 modulates effector T-cell activity in peripheral tissues and in the tumour microenvironment (TME), whereas CTLA-4 prevents T-cell activation [15]. A recent study by Polverino and coworkers analysing PD-L1 expression in several cell types of lung structure and inflammatory cells from two cohorts of COPD and NSCLC patients demonstrated that both NSCLC patients and patients with a mild stage of COPD expressed the highest levels of PD-L1 in alveoli, bronchioles, and vessels compared to patients with severe COPD stages [19], further corroborating the existence of a strong link between the two pathological conditions.

### 2.1. The Biological Pathway of CTLA4

CTLA4 is expressed on T cells with the primary aim of regulating the initial CD8+ cell activation. As widely described, CTLA4 can act through both direct and indirect inhibition of T-cell proliferation [15]. The former is mediated by the linking with its renowned ligands CD80 (or B7.1) and CD86 (or B7.2), probably by downstream activation of two protein phosphatases, PTPN11 and PP2A, thus counterbalancing the activation of protein kinases induced by T-cell antigen receptor (TCR) and CD28 stimulation [18,20,21]. CTLA4 shows a higher affinity for CD80 and CD86; consequently, its expression on the membrane surface of CD8+ T cells antagonises CD28 by competing for the binding sites and segregating them, thus indirectly inhibiting T-cell activation [18,20,21]. Moreover, a process of CTLA-4-mediated trans-endocytosis of CD80 and CD86 by dendritic cells to be degraded has been described [17]. Notably, another important role of CTLA-4 expressed on CD4+ T cells is the immunomodulation of the two main subtypes, TH-1 and T regulatory (Treg) cells, through downregulation of the former and upregulation of the latter, leading in both cases to immunosuppression [15]. Consequently, ICI directed against CTLA-4, such as Ipilimumab and Tremelimumab, can consistently shape the immune response in the TME.

### 2.2. The Biological Pathway of PD-1/PD-L1

PD-L1 can largely be found in T cells and in activated B cells acting as an immunosuppressive signal, through the stimulation of receptor PD-1 leading to CD8+ T-cell apoptosis [15]. Importantly, the TME itself can induce PD-L1 transcription at different levels, transcription, post-transcription, and post-translation, by numerous factors, including inflammatory stimuli and oncogenic pathways, indirectly inducing immune response downregulation [22,23,24]. Among the inflammation-related transcription factors regulating PDL-1 overexpression is nuclear factor kappa-B (NF-κB), which mediates PD-L1 overexpression-induced interferon-γ (IFN-γ) [25]. IFN-γ, in turn, activates JAK-STAT signalling [26,27], thus upregulating the expression of interferon-responsive factors (IRFs) [22]. Besides IFN-γ, other inflammatory transcription factors are IL-6, related to MEK-ERK signalling [28], and tumour necrosis factor-α (TNF-α), all involved in NF-κB pathway activation [25,29].

The role of the receptor PD-1 is to limit the over-activation of CD8+ T cells during prolonged inflammatory responses such as infections or autoimmune disease, resulting in a balance between the efforts in clearing the pathogens and the immune-related damage [30]. The protein expression of PD-1 increases when activated CD8+ T cells gather in the tumour-infiltrating lymphocytes (TILs); therefore, its expression represents an important mechanism for immune escaping, inducing anergy and CD8+ exhaustion in chronic viral infections [31]. Analogously to CTLA4, PD-1 is also expressed on Treg cells, increasing their proliferation and consequently the suppression of the immune response [32]. In the literature, two main ligands for the PD-1 receptor are described, namely PD-L1, also known as B7-H1, and PD-L2, alternatively known as B7-DC23 [33]. Notably, chronic inflammation such as chronic infection and neoplasms can persistently increase PD-1/PD-L1 expression, inducing tolerance and, consequently, tumour survival. This setting has been elucidated in chronic viral infections both in mice and in humans and, apparently, is partially reversible by the use of ICIs [34].

### 2.3. Role of Tumour Microenvironment and Myeloid Cells

The TME is a complex ecosystem of cells and molecules that surround and intermingle with tumour cells interacting with immune elements, both adaptive (including CD8+ T cells, Th1 cells, Th2 cells, Th17 cells and B cells) and innate (including neutrophils, macrophages, and natural killer cells), fibroblasts, and the niche [35]. In this environment, cancer cells can evade immunity, become resistant to ICIs, and, eventually, metastasise [36,37]. In particular, myeloid cells, a heterogeneous subtype of innate immune cells such as macrophages and dendritic cells (DCs), have a significant ability to modulate CD8+ T-cell responses involving immune escaping, survival, and cancer growth [38]. Alveolar and tissue-resident macrophages play a central role in COPD and NSCLC. In COPD, macrophages contribute to chronic inflammation, release pro-inflammatory cytokines (IL-1β, TNF-α), and secrete proteases, leading to tissue damage. Tumour-associated macrophages (TAMs), from circulating monocytes, promote tumour growth and immunosuppression in NSCLC through the release of immunosuppressive factors like IL-10 and TGF-β. They can exhibit pro-inflammatory (M1) as well as anti-inflammatory/immunosuppressive (M2) phenotypes, both of which are altered in COPD [39]. Interestingly, tumour tissues exhibited low levels of M1 macrophages and M1/M2 ratios, while the concomitant presence of COPD significantly enhanced the M1/M2 ratio, although this did not impact patients’ survival [40]. Within the TME, DCs can be functionally impaired, leading to reduced anti-tumour immune responses. It has been recently demonstrated that the lung tissues of patients with COPD have a peculiar immune cell distribution, containing a high number of resting NK cells, activated DCs, and neutrophils, and a lower fraction of follicular T helper cells and resting DCs [41,42]. As a result of this intense traffic, any modification in the TME can hypothetically affect the efficacy of ICIs. Among the immune checkpoint molecules are some metabolic enzymes like the indoleamine 2,3-dioxygenase (IDO), an effector expressed by tumour cells and myeloid cells of the tumour niche, and arginase, an amino acid produced by immune-suppressor cells by the myeloid line [43,44]. These enzymes are delivered to the TME and can inhibit immune responses through the local degradation of amino acids used for T-lymphocyte biosynthesis, like tryptophan, or through the action of specific natural ligands, thus altering lymphocyte functions in both COPD and NSCLC [44]. The IDO expression is upregulated in both these pathologies, leading to T-cell inhibition and immune escape [43,44]. Table 1 reports the immune cells predominantly infiltrating the lung TME in NSCLC, COPD, and both (Table 1).

## 3. COPD Affects the TME by Altering the Response of NSCLC to ICIs

In the past ten years, the immune microenvironment, a key player in the TME, has been linked with NSCLC occurrence and disease progression [54]. With COPD being an inflammatory disease, the immune microenvironment could be reprogrammed by the inflammation itself, thus increasing the risk of lung cancer development and tumour progression [55] and reducing the efficacy of therapies [56]. The development of COPD could be affected by PD-1/PD-L1 protein expression through their interaction with the TME and various factors, such as apoptosis of T lymphocytes, alteration and modulation of immune checkpoint proteins, and the modulating effect of released cytokines on immune and tumour cells. Presumably, sharing a similar pathogenesis could justify the better benefits observed with the use of PD-1/PD-L1 inhibitors [57,58,59]. Furthermore, a novel immune cell subtype, myeloid-derived suppressive cells (MDSCs), a heterogeneous group of cells with significant immunosuppressive activity that promote tumour growth by suppressing effector T-cell function, could mediate tumour immune escape [60].

### 3.1. Common Pathways in NSCLC and COPD Affect PD-1/PD-L1 Expression and ICI Efficacy

Patients affected by COPD coexisting with NSCLC showed variable responses to ICIs, suggesting that the COPD-related TME could modulate the treatment efficacy [5]. Approximately 35–70% of lung cancer patients are also diagnosed with COPD [8,54,61], the frequency of COPD in the squamous cell histotype being higher than in adenocarcinoma [54,56]. Genes linking COPD and lung cancer were analysed in several studies with next-generation sequencing (NGS) techniques [62,63]. The addition of artificial intelligence and bioinformatics is currently used for profiling common pathological mechanisms and possible prognostic or predictive genomic biomarkers [64]. Pre-clinical studies found that the expression of the Zinc Finger transcription factor ZNF143 was upregulated in NSCLC, both in adenocarcinoma and squamous cell histotype samples, and its increased expression was significantly associated with advanced TNM stages, although survival analysis failed to demonstrate its role as a prognosticator [54]. Interestingly, the ZNF143 protein was associated with increased levels of CD8+ TILs and high levels of tumour mutational burden (TMB) both in patients with COPD and NSCLC, thus highlighting its potential role as a predictive biomarker for good responses to ICI when a patient is affected by both diseases [54]. Indeed, high levels of CD8+ TILs and TMB associated with NSCLC were previously described as a general marker for good responses to immunotherapy and PD-L1 blockade [5,6,65,66,67,68]. Generally, cells at risk of malignant transformation frequently harbour genetic mutations in important onco-suppressor genes such as TP53 and CDKN2A [69,70]. T helper cells of type 17 (Th-17) were found to be related to COPD progression, whereas interleukin-17 receptor A deficiency, such as in congenital deficient mice, is protective against smoke-induced emphysema [71]. Furthermore, pieces of evidence from pre-clinical models of lung cancer, such as genetically modified mice, were gathered for IL-17, showing an increase in the inflammatory response and the induction of lung tumour progression and growth [72]. Presently, it is yet unclear if Th-17’s pro-tumour effect could be related to COPD metabolic and genetic alterations or if the latter are just linked to cigarette smoking activity [5]. Defining the right correlation between cause and effect could be hard, and the relationship between COPD and NSCLC could be a false relationship, where smoking habits could represent a common risk factor that justifies disease progression. However, CD8+ T-cell status can be influenced by COPD co-presence as well as PD-1 status, significantly associated with tumour progression. Biton and colleagues observed that patients with NSCLC and COPD showed an upregulation of PD-1 and T-cell immunoglobulin and mucin domain-containing protein 3 (TIM3) in CD81+ cell lines [6]. Indeed, TIM-3 expression is often associated with chronic immune activation and exhaustion [73]. The binding of TIM-3 to its ligands, including galectin-9 and carcinoembryonic antigen-related cell adhesion molecule 1 (CEACAM1), can negatively regulate immune responses [73]. Engagement of TIM-3 on CD8+ T cells also leads to impaired T-cell proliferation, cytokine production, and cytotoxicity [74].

### 3.2. COPD Comorbidity Enhances TILs and Th-1 Response in Lung Cancer Patients

One of the first features taken into account as possible predictors was TILs [75]. Indeed, some authors showed how COPD strongly affects the immune microenvironment of NSCLC, and CD8+ TILs have been identified as the most affected population, to the extent that CD8+ TIL exhaustion was correlated with COPD severity [6]. On the other hand, TILs in the TME were shown to become less active, or even inactive, failing to clear the tumour cells. This phenomenon is called “exhaustion” and it is probably caused by chronic inflammation and prolonged exposure to tumour antigens, inflammatory cells, and inflammatory cytokines [16], leading to immunosuppressive epigenetic modifications in T cells [76]. However, in vivo and in vitro studies have suggested that this phenomenon is reversible when ICIs are used, through PD-1/PD-L1 inhibition, restoring the immune response, and warranting better survival in different subgroups [16,75,77]. Patients with both COPD and NSCLC show a heterogeneous differentiation of the CD4+ sub-phenotypes, with Th-1 predominance being reported in some studies [5] or, conversely, enriched Treg immunity [57]. Indeed, a relative predominance of T helper 1 cytokines and M1 macrophages has been observed in patients with COPD and NSCLC, compared to NSCLC alone, indicating the presence of a stronger pro-inflammatory pattern in the former condition [40]. Some authors detected a strong inverse correlation between neutrophil (CD66b+) count and CD8+ T-cell count [78,79]. Chen et al. described a gene expression profiling technique for estimating the immune cell content [80]. The CIBERSORT method investigated 18,000 patients affected by more than 30 cancer types, describing the immune cell composition in the population with COPD-NSCLC. They found that neutrophil count could predict mortality better than other inflammatory cells in all histologies but mostly in lung adenocarcinoma [81,82]. Additionally, smoking habits could predict a more favourable response in patients with NSCLC treated with anti-PD-1 drugs, probably due to the increasing tumour mutational burden induced by carcinogens released by cigarette combustion [83]. In a subset of patients with COPD, the interferon-gamma (IFN-γ) signature, harbouring a Th-1 response, has been shown to positively predict the response to anti-PD-1 treatments [84], with a rising count of CD8+ and CD4+ cells in these patients. Furthermore, increased Th17 content was associated with high levels of PD-1 expression in the same lung tissues of patients with COPD [5], who had longer progression-free survival if treated with ICIs. This suggests that COPD could indirectly represent a good predictor of responses to immunotherapy. Supporting this finding, a clinical study showed that high levels of Th-1 lymphocytes could enhance ICI activity, predicting better survival after chemotherapy, whereas an increased number of Th2 cells showed a decrease in immune response and predicted worse results [85].

### 3.3. Other Predictive Biomarkers Evaluated in NSCLC Patients with COPD

Other molecules such as PD-1 and TIM3 could induce T-cell depletion and were thus considered markers of T-cell exhaustion and unresponsiveness to ICIs in NSCLC, in particular in patients with comorbid COPD [5,6]. Indeed, NSCLC patients benefit more from anti-PD-1 therapy in the presence of COPD than patients presenting with only lung cancer, showing significantly higher survival rates. Notably, this suggests a better effectiveness of PD-1 blockers in reactivating the CD8+ T-cell response in the cell subpopulation characterised by TIL exhaustion [6].

Th-17 cells are also promising biomarkers and have an opposite role compared to Treg cells in regulating the immune system, inhibiting each other to maintain dynamic balance. Notably, this balance is also pivotal in modulating the anti-cancer response and autoimmune disorders [86]. In particular, in vitro investigations found that Th17 cells could promote T-cell recruitment in the tumour site and raise the CD8+ TILs [5,71,72]. This balance can be altered in the presence of COPD influencing Treg and Th17 cell counts and efficacy, thus affecting ICI outcomes [87]. Furthermore, since an imbalance between Treg and Th17 cells plays an important role in the immune response of cells, particularly in autoimmune diseases, but also in the development of malignant tumours [88,89], in patients with advanced NSCLC and COPD, the study of Treg and Th17 indicators could set the basis for a new combined strategy for tumour treatment [87].

Vascular endothelial growth factor (VEGF) was shown to be an interesting biomarker due to its key regulatory nature in angiogenesis being overexpressed in both lung diseases. VEGF production in COPD represents a mechanism to compensate for impaired lung function and hypoxia, but it has a role in attenuating the immunity vigilance when COPD coexists with advanced NSCLC [90]. VEGF was found to be increased in patients treated with Bevacizumab, correlating with good prognosis [91]. On this matter, some studies explained how raising levels of VEGF in COPD patients with NSCLC resulted in higher circulating levels of monocytic MDSCs (M-MDSCs) and lower levels of granulocytic MDSCs (G-MDSCs), leading to increased survival of NSCLC patients [90]. Conversely, M-MDSC and G-MDSC levels were decreased in NSCLC patients without COPD. Moreover, while a linear correlation between VEGF and M-MDSCs was observed in NSCLC, an inverse correlation was described in COPD [90].

Interleukins (ILs) and their receptors have been investigated as potential biomarkers to monitor ICIs’ effectiveness in patients with NSCLC and COPD. A recent meta-analysis evaluating the role of all cytokines for which critical roles in the immune process have been established [92] indicated that raised levels of IL-8 and IL-2 receptor (IL-2R) during ICI treatment could predict a poor response [7]. Accordingly, lower values of IL-8 were found at baseline in patients affected by NSCLC and COPD, predicting longer overall survival (OS) and progression-free survival (PFS) in some studies [7,93], probably acting through a reduction in oxidative stress [94], epithelial–mesenchymal transition [95], angiogenesis [96], immunosuppression, and gathering MDSCs [96]. On the other hand, the chronic pulmonary inflammation created by COPD produces immunosuppressive cytokines such as IL-10 that contribute to enhanced tumorigenesis and cancer development [58].

Epigenetic modifications, such as DNA methylation/demethylation and histone acetylation, can alter the gene expression patterns in both COPD and NSCLC [97]. Aberrant epigenetic marks can affect the function of genes involved in inflammation, cell cycle regulation, and DNA repair, among others. These changes can create a permissive environment for cancer development and compromise the response to therapy. DNA methylation and its modulatory mechanisms have long been investigated as potential biomarkers for ICIs’ effectiveness [98]. Previous studies showed that tumour DNA methylation profiles could significantly differentiate the immune cell proportions in the TME, including TILs [99], whereas the methylation of promoters CTLA4, LAG3, and PD-L1 was largely associated with longer survival and a longer duration of response when immunotherapy was used [100]. Some studies have shed light on the role of DNA methylation in COPD development and as a potential biomarker in patients with NSCLC and COPD [100,101,102], and the profiling of DNA methylation signatures in NSCLC and COPD patients revealed important differences between patients with or without COPD [101,102]. Notably, a recent epigenome-wide association study clarified the link between the co-presence of COPD and NSCLC and the methylation status of the genome. Indeed, the quantification of methylation and the consequent gene repression was found to be highest in patients with co-presence of COPD and NSCLC, while COPD and NSCLC alone were described to be enriched in methylated genes in second and third place, respectively [103]. In particular, the authors described an important relationship between immune checkpoint inhibitor sensitivity and gene repression by miRNA activity when investigating the epigenetic causes of resistance to immunotherapy.

Other post-translational modifications altering gene expression patterns in patients with NSCLC, COPD, or both are reported in Table 2.

The major histocompatibility complex (MHC), DP alpha 1 (DPA1), also known as HLA-DPA1 play an important role in antigen presentation in cells related to immune response regulation. HLA-DPA1 is abnormally expressed in lung cancer tissues, thus correlating with cancer progression, OS, and DSS, and may be regarded as a potential prognostic marker [113]. Furthermore, previous reports have indicated HLA-DRA as a promising biomarker that, being associated with an inflamed TME, might guide immunotherapy in NSCLC [114].

In line with this evidence and as NSCLC might block the expression of HLA class I molecules to escape immune surveillance, potential immunomodulatory and MHC-encoding genes overexpressed in NSCLC and COPD have been studied as promising biomarkers of responses to immunotherapy [113,114,115,116,117,118].

## 4. Observational Studies Investigating COPD’s Influence on NSCLC Patients Treated with ICIs

A high incidence of COPD in NSCLC patients is widely described in the literature [119], suggesting a significant incidence of both comorbidities in the same patients. Indeed, in patients with advanced NSCLC, some authors described a prevalence of COPD ranging from 39% to 50% [119,120]. Consequently, ever-growing attention has been paid to this cohort, leading to special recommendations by the Global Initiative for Chronic Obstructive Lung Disease (GOLD) that guide clinicians on the management of patients with COPD, stating that concomitant chronic diseases, including lung cancer, should be actively sought and appropriately treated as they can independently influence both mortality and hospitalisations [120]. Several studies that investigated patients affected by NSCLC and a moderate-to-severe COPD comorbidity showed prolonged PFS and OS [5,7,68,121,122] and higher ORR [101,123] when compared to patients without COPD, and better survival and responses were also observed in patients with only mild COPD [124] (Table 3).

Notable, despite OS not significantly differing between COPD and non-COPD patients, it became significant when the patients were stratified for smoking status (being higher in current smokers than in previous smokers) [5,6,68,123]. Interestingly, some authors showed a significant correlation between several cytokines and survival in this category of patients. In particular, Zhang et al. investigated how IL-6, IL-8, and IL-10 could predict different responses to ICIs [125]. In a univariate analysis, each predictive marker proved to be significantly correlated with median PFS, such as the GOLD score for COPD (HR = 1.7, *p* = 0.034) and the plasma levels of IL-6 (HR = 1.001, *p* = 0.003), IL-8 (HR = 1.014, *p* = 0.005), and IL-10 (HR = 1.049, *p* = 0.014). However, in a multivariate analysis, only IL-6 remained significantly correlated with median PFS (HR = 1.001, *p* = 0.007) [125].

## 5. SAFETY and Interstitial Lung Disease (ILD)

Interstitial Lung Disease (ILD) is a chronic inflammatory condition often associated with ICI treatments, and it causes a high rate of immunotherapy discontinuation and significant mortality and morbidity in NSCLC patients [126,127]. A recent network meta-analysis showed that the incidence of ILD in NSCLC was about 3–4% overall including all types of immunotherapy. However, the incidence was higher for PD-1 inhibitors than for PD-L1 inhibitors [127]. In addition, with COPD being a chronic inflammatory disease that often, as already stated, coexists with NSCLC, its presence represents a further potential risk factor for ILD [128].

Indeed, a prediction model for ILD used the presence/absence of COPD as one of the predictors to estimate the incidence of ILD together with other factors (age, smoking, chest radiotherapy) for decision making regarding ICI selection [126,129]. Lin et al. evaluated COPD as a potential predictive marker in NSCLC patients by stratifying three different subgroups based on PD-L1 expression (PD-L1 < 1%, PD-L1 1–49% and PD-L1 ≥ 50%), failing to find significant differences among these groups [58]. For this reason, the investigation of new predictive biomarkers or prediction scores is of utmost importance.

Independently of PD-1/PD-L1 levels of expression, ICIs used as monotherapy showed higher risks of both grade 1–5 immune-related adverse events (irAEs) (OR 2.14, 95% CI: 1.12–4.80) and grade 3–5 irAEs (OR 3.03, 95% CI: 1.49–6.69) than when used in combination with chemotherapy. Analogously, double-ICI combinations significantly increased the risk of any grade of irAE (OR 3.86, 95% CI: 1.69–9.89) and, dramatically, that of grade 3–5 irAEs (OR 5.12, 95% CI 2.01–13.68). Of note, no significant difference was observed between ICI-based doublets and monotherapies in the occurrence of irAEs of any grade (OR 1.85, 0.91–3.37) or grade 3–5 (OR 1.65, 95% CI: 0.81–3.37). Furthermore, PD-1 inhibitors showed more toxicity than PD-L1 inhibitors, with a higher risk of irAEs of any grade (OR 2.56, 95% CI: 1.12–6.60) [127].

In a study by Zhang et al., 30 out of 99 patients experienced irAEs. Three subgroups of patients were identified based on the severity of COPD (mild, moderate, or severe) and a control group with no comorbidities was considered for comparison [125]. The most-described symptoms of irAEs were enteritis and diarrhoea (4.2–8.0%), ILD (n1 0.0–28.0%), and immune-related hepatitis (8.0%, only in the group with severe COPD) [125]. When predictors were studied in a univariate analysis, the authors observed that the GOLD score for COPD (HR 2.0, *p* = 0.003), IL-6 (HR 1.03, *p* = 0.007), IL-8 (HR 1.03, *p* = 0.022), and IL-10 (HR 1.35, *p* = 0.006) positively correlated with the incidence of irAEs, although in a multivariate analysis, only the GOLD score remained significantly associated with irAEs (HR 1.8, *p* = 0.037) [125]. Immune-related ILD (irILD) was investigated in a large cohort of patients treated with different regimens of ICIs, both in monotherapy and in doublets [130]. In total, 40% of irILDs occurred in NSCLC patients and, in particular, 11.5% in those with asthma, while only 4.34% of irILDs occurred in non-asthmatic patients. Of note, all irILDs occurred in the asthmatic/COPD group with severe symptoms (irILD grade 3–5), women, and heavy smokers [130]. In a recently published study on PD-1/PD-L1 inhibitors used in a neoadjuvant setting in combination with chemotherapy, 20% of patients in the COPD group and 20.5% of those in the non-COPD group had grade 3 or 4 treatment-related AEs, mainly indicative of haematological toxicity [121].

## 6. Discussion

The activity of ICIs in NSCLC patients with COPD has generally shown good outcomes, although controversial data with worse overall response rates have been reported, in particular, for patients affected by both NSCLC and severe COPD (8.0%) rather than mild-to-moderate COPD (37.5–35.5%) or an absence of comorbidities (31.6%) [125]. Notably, a slight increasing trend of ORR in NSCLC patients with mild-to-moderate COPD was observed, although not significant (*p* = 0.589), whereas a dramatic decrease in response rates was assessed in severe COPD (*p* = 0.009) [125]. These results suggest a negative interaction of COPD with the immune system, although possible alternative explanations might be an increase in bronchitis/pneumonitis, hospitalisations, poor compliance, and the high rate of side effects caused by immune stimulation in this patient subpopulation. However, no role for COPD as a prognostic or predictive factor was found in patients with different PD-L1 expression (PD-L1 < 1%, between 1 and 49% and > 50%), better explaining the efficacy or the lack of response in these subgroups of patients [58]. Further research on the possible relationship between PD-L1 expression and ICIs is needed.

The effect of ICIs on NSCLC patients with emphysema was also investigated. PFS and OS were demonstrated to be superior for mild-to-moderate COPD patients (19.5 months and 6.6 months, respectively) than for patients with no emphysema (11.0 and 2.7 months, respectively) with differences that were statistically significant (*p* = 0.032 and *p* < 0.001, respectively) [68]. It is likely that worse survival in patients with severe COPD is linked to poor prognosis, poor compliance, and the need for hospitalisation, whereas the positive trend in ORR and PFS for mild-to-moderate COPD suggests a possible prognostic/predictive role of COPD in NSCLC patients. Some authors described a significant improvement in OS in NSCLC patients affected by COPD, but this advantage was lost when COPD status coexisted with smoking habits (>30 packs/year) [68]. An even higher impact of COPD has been reported in a multivariate analysis adjusted for several other factors, including smoking habits [6,124]. Smoking habits are renowned for being a good predictive factor for the response to ICIs [131,132]; consequently, it is very likely that interference in COPD patients could affect outcomes in NSCLC patients with COPD. Indeed, cigarette smoking is a pathogenic trigger for both COPD and NSCLC, and COPD prognosis strongly depends on smoking habits [8]. Studies investigating this link with interaction tests might be useful to assess the strength of this interference.

Among other predictive markers, IL-6 proved to be significant in a multivariate analysis adjusting for known factors (i.e., smoking, COPD, age, or tumour histotype) [124]. Unluckily, this marker failed to be clinically meaningful (HR = 1.001) [125], whereas baseline and on-treatment changes in IL-8 and IL-2R proved to be useful survival predictors in patients receiving immunotherapy [7]. Targeting the Treg/Th17 balance for therapeutic purposes in addition to ICIs might represent a useful tool for future lung cancer treatment strategies [87,133] in order to suppress Treg cells and enhance Th17 cells.

Other than efficacy, safety and potential adverse events should also be carefully considered when a treatment based on ICIs has to be chosen in NSCLC patients with COPD. Indeed, ICI-based drugs were found to increase the risk of immune-related pneumonitis (IRP) of any grade and, importantly, of grades 3–5. On this behalf, when ICIs are used in monotherapies, they display a doubled risk of IRP of any grade and a tripled risk for grade 3–5 IRP in comparison with the same drugs used in association with chemotherapies [127]. This is probably due to a myelotoxic effect driven by antiblastic drugs, eventually influencing the immune system. A dramatic increase in IRP was observed when ICI-based doublets were compared with ICIs + chemotherapy, with risks almost 4-fold higher for any grade of immune toxicities and 5-fold higher for IRP grade 3–5 [127], caused by the use of double blockades or any additional pathway like the more myelotoxic anti-CTLA4 drugs. However, when compared with ICI monotherapies, doublets showed an 85% higher risk of any-grade irAEs and a 65% additional risk for grades 1–5, although this did not reach statistical significance. Additionally, PD-1 inhibitors seemed to cause an increased risk for IRPs (OR 2.5) compared to PD-L1 inhibitors [127], suggesting that drugs such as Atezolizumab or avelumab should be preferred in patients with COPD as a comorbidity. The incidence of irILD can be predicted in the presence of lung conditions, such as asthma and COPD, pre-existing before the initiation of ICIs; unfortunately, data on this issue are still poor [128]. However, Pembrolizumab showed, at the time of FDA approval, an incidence of 3.5% for immune-related pneumonitis, with 2.2% requiring drug discontinuation. Notably, patients with a medical history positive for asthma or COPD suffered ILD more than patients free from these comorbidities (5.4% vs. 3.1%). Nonetheless, biases and unbalanced proportions limited the speculations on the predictivity of this marker. Indeed, no direct comparison with a statistical test was made, COPD and asthma were grouped together despite being driven by different factors, and the experimental and control arms were not stratified for these comorbidities. Nonetheless, the association between pneumonitis and asthma was confirmed by a small study linking the incidence of irILD grades 3–5 with symptomatic asthma, thus identifying a 3-fold increased risk when compared to non-asthmatic patients (11.5 vs 4.34%) [130]. A similar study by Atchley et al., investigating the incidence of irILD in NSCLC patients treated with Nivolumab, Pembrolizumab, or combination of Ipilimumab + Nivolumab, described a 15-fold increased risk when a history of ILD was present, a 3-fold increased risk when patients were also affected by COPD, and a 7-fold increased risk when patients had fibrosis at baseline. However, despite being interesting, the small size of these two studies and the heterogeneity of the regimens (monotherapies were flanked with doublets in the same cohort) also limit the robustness of the results. Other chronic inflammations of the lung were described to raise the irILD risk. A retrospective study, investigating the delivery of Nivolumab in 216 NSCLC patients, revealed that irILD occurred more often in patients with an ILD diagnosis at baseline, both for any grade (31% vs. 12%, *p* = 0.014) and irILD grade 3–5 (19% vs. 5%, *p* = 0.022) [134]. A similar study, investigating irILD incidence in NSCLC patients also affected by lung fibrosis at the time of treatment with Nivolumab, recorded increased rates of immune-related pneumonitis [135]. Although these data could prove useful, they are not exactly driven by COPD or asthmatic inflammation and, consequently, suffer from indirectness bias, making them difficult to generalise.

## 7. Conclusions

In conclusion, patients with NSCLC and affected by COPD (or emphysema) achieved better outcomes when treated with immune checkpoint inhibitors. Significantly better results are observed for PFS and ORR in patients affected by COPD, with current smokers benefitting the most, whereas for OS outcomes, no statistical difference is observed between the two groups. However, when patients are stratified for current smoking habits, OS outcomes are better for patients with coexisting COPD comorbidity, suggesting that an inflammatory status maintained by smoking would improve immunotherapy at the cost of increased autoimmune toxicity. Definitive results could come from prospective correlative studies whose design would include clinical and molecular prognostic and toxicity-predictive factors.

## Figures and Tables

**Figure 1 cancers-16-01251-f001:**
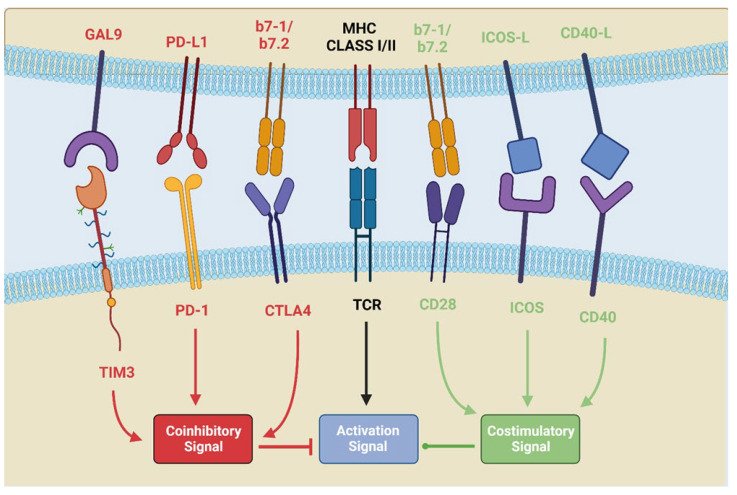
Balance of costimulatory and co-inhibitory signals acting on the main CD8+ maturation signal. Image created for *Cancers* journal by R.R. with BioRender.com (accessed on 29 December 2023).

**Table 1 cancers-16-01251-t001:** Immune cells predominantly infiltrating the lung tissue microenvironment in patients with NSCLC, COPD, and both diseases.

Immune Cell	NSCLC [45,46,47,48,49,50,51,52,53]	COPD [42,48,51]	NLCSC and COPD [48,53]
Plasma cells	x		
M2 macrophages	x		
M0/M1 macrophages	x	x	x
CD8 T cells	x	x	x
Resting CD4 T cells	x		
CD4 T cells		x	
CD4 Treg cells	x		x
Mast cells	x		
Memory B cells	x		
Dendritic cells		x	
Neutrophils	x	x	x
NK	x		

NSCLC: non-small cell lung cancer; COPD: chronic obstructive pulmonary disease; x indicates presence.

**Table 2 cancers-16-01251-t002:** List of described de-regulated post-transcriptional modulators in NSCLC, COPD, and both.

Post-Transcriptional Regulations	Involved Molecule	NSCLC	COPD	NSCLC and COPD
lnc-RNA	SCAL1 [104,105]		•	•
MALAT1 [106]	•		
UCA1 [107]	•		
HOTAIR [108,109]	•		
H19 [110]	•		•
miRNA	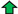 miR-675 [110]	•		•
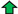 miR-33a-5p [111]	•	•	•
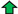 miR 149-3p [111]	•		
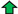 miR 197-3p [111]	•		
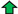 miR 199a-5p [111]	•		•
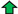 miR 320a-3p [111]	•		•
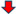 miR-34a-5p [111]	•	•	
circ-RNA	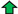 circ_0047921 [112]		•	
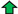 circ_0056285 [112]			•
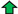 circ_0007761 [112]			•

Lnc-RNA: long non-coding Ribonucleic Acid; miRNA: micro-RNA; circ-RNA: circular RNA; • indicates presence.

**Table 3 cancers-16-01251-t003:** Studies investigating outcomes of NSCLC patients with and without concomitant COPD.

Authors	Year	Type of Study	Sample Size	Type of Drug	Name of Drug	Mean Age (±SD)	COPD			No COPD			HR OS	HR PFS
							OS	PFS	ORR	OS	PFS	ORR		
Mark et al. [5]	2018	Retrospective	72	Anti-PD-1 Anti-PD-L1	Nivolumab Pembrolizumab Atezolizumab	66.4 ± 9.1	359 d *	153 d	-	145 d *	54 d	-	Higher survival in COPD and smoker patients (*p* = 0.0350)	0.58 (*p* = 0.033)
Biton et al. [6]	2018	Retrospective	39	Anti-PD-1	Nivolumab	64 ± 9	250 d **	100 d **	-	450 d **	150 d **	-	Higher survival in COPD and smoker patients	Higher survival and disease response in COPD and smoker patients
Suzuki et al. [101]	2019	Retrospective	229	NR	NR	NR	30 ms	-	-	36 ms	-	-	Higher survival when genetic factors were stratified	-
Shin et al. [124]	2019	Retrospective	133	Anti-PD-1	Pembrolizumab	63	5 ms **	2 ms **	38.2%	8 ms **	6 ms **	20.5%	0.51 ^#^ (*p* < 0.001)	0.61 ^#^ (*p* < 0.001)
Takamori et al. [123] ^§^	2020	Retrospective	257	Anti-PD-1 Anti-PD-L1	Nivolumab Pembrolizumab Atezolizumab	66	14 ms	6 ms	-	28 ms	3 ms	-	0.526 (*p* < 0.0001)	0.672 (*p* = 0.0006)
Takayama et al. [68] ^§^	2021	Retrospective	153	Anti-PD-1 Anti-PD-L1	Nivolumab Pembrolizumab Atezolizumab	68 ± 9.5	19.5 ms	6.6 ms	32.4%	11.6 ms	2.7	15.9%	0.58 (*p* = 0.03)	0.47 (*p* < 0.001)
Zhou et al. [7]	2021	Retrospective	156	Anti-PD-1 Anti-PD-L1	NR	NR	510 d	316 d	-	Not reached	186 d	-	0.56 ^#^ (*p* = 0.018)	0.56 ^#^ (*p* = 0.034)
Noda et al. [121] ^§^	2022	Retrospective	56	Anti-PD-1 Anti-PD-L1	Nivolumab Pembrolizumab Atezolizumab	70	20.6 ms	6.5 ms	34.1%	10.8 ms	2.3 ms	6.7%	0.36 (*p* = 0.004)	0.30 (*p* = 0.005)
Dong et al. [122]	2024	Retrospective	74	Anti-PD-1 Anti-PD-L1	Nivolumab Pembrolizumab Atezolizumab Durvalumab	63.87 ± 5.87	-	Not reached	70%	-	17 ms	63.6%	-	χ2 = 6.247 (*p* = 0.012)

* significant OS was reported for current smoker patients. ** Data retrieved from the graphs (approximated at breaks of 50 days). ^#^ The author adjusted the HR values for histology, Stage IV at diagnosis, and number of previous lines of chemotherapy. ^§^ In these studies, emphysema was investigated as a prognostic factor. d = days, ms = months, NR = not reported, SD = standard deviation.

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
