# Peer review of "Effectiveness of Immunotherapy in Non-Small Cell Lung Cancer Patients with a Diagnosis of COPD: Is This a Hidden Prognosticator for Survival and a Risk Factor for Immune-Related Adverse Events?"

_cancers, 2024, doi:10.3390/cancers16071251_

Round 1
Reviewer 1 Report
Comments and Suggestions for Authors
1. Please include a table indicating the types of immune cells predominantly infiltrated in the lung tissue microenvironment in NSCLC patients, patients suffering from COPD, and in those with both NSCLC and COPD.
2. Please include a table indicating the immunomodulatory and MHC encoding genes overexpressed or downregulated predominately in the lung tissue microenvironment in patients suffering either from NSCLC, or COPD only, or from both.
3. Apart from methylation and acetylation expand on other posttranslational modifications altering gene expression patterns in COPD and NSCLC separately or in combination of the two. Please present these genes in a table.
4. Page 4, lines 132, 133: “Importantly, TME itself can induce PD-L1 transcription, indirectly inducing immune response downregulation [22,23].” Please provide further insight in the molecular mechanisms involved specifying the transcription factors contributing to PD-L1 gene overexpression in the TME.
5. Page 5, lines 208-210: “Generally, cells at risk of malignant transformation frequently harbour genetic mutations in important onco-suppressor genes such as TP53 and CDK2N2A [53,54].” Please provide the gene symbols in Homo sapiens (CDKN2A instead of CDK2N2A)
6. Page 1, lines 34, 35: “In the present review, we tried to understand the interplay between the two pathways…” Authors should indicate the two pathways they refer to.
7. Page 7, lines 314-318: “Indeed, the quantification of methylation and the consequent gene repression was found to be maximum in patients with co-presence of COPD and NSCLC, while COPD and NSCLC alone were described to be enriched in methylated genes in second and third place, respectively.” This information is interesting, but the question is not the number of genes methylated in each case but whether immune checkpoint molecules, pro-inflammatory or anti-inflammatory genes were preferentially modulated in each case.
8. Page 7, lines 325-327: “Consequently, an ever-growing attention has been paid to this cohort, leading to special recommendations by the Global Initiative for Chronic Obstructive Lung Disease (GOLD) guidelines [87].” Please specify what these guidelines are and how they have been influenced by the scientific findings described in this review.
9. Please rephrase the following sentences as their meaning is not clear:
· Page 5, lines 230, 231: “On this purpose, some authors showed how the depletion of CD8+ lymphocytes in TILs correlated with COPD severity [6].”
· Page 6, lines 234, 235: “…and the durable exposure to tumour antigens and co-inhibitory signals [16]…” Please rephrase the meaning of the sentence is not clear.
· Page 6, lines 253-255: “Furthermore, Th-1 response was linked with high levels of PD-1 expression in the lung tissues of patients with COPD5, suggesting that COPD could indirectly predict a response in patients treated with ICI.”
· Page 6, line 264: “…much more when combined with COPD…” This sentence does not make sense.
· Page 7, lines 268, 270: “Th-17 cells are also promising biomarkers and have an opposite role compared to T-reg cells in regulating the immune system, balancing their activating against their inactivating stimulation.”
· Page 7, lines 322-324: “Indeed, some authors described 69% of patients with COPD (about 39%), emphysema (about 59%) or both, in the NSCLC patients followed while others reported an incidence of COPD in 50% patients with advanced NSCLC.”
· Page 3, lines 121-123: “Moreover, an active sequestration of CD80 and CD86 by the antigen-presenting cell (APC) membrane surface, has been described17.” [17]
Comments on the Quality of English LanguageFew sentences, indicated above, should be rephrased to clarify their meaning.
Author Response
We thank the Reviewer for her/his comments. A point-by-point answer is uploaded as a separate file.

Reviewer 2 Report
Comments and Suggestions for Authors
Comments:
The manuscript describes "Effectiveness of immunotherapy in patients with non-small cell lung cancer diagnosed with chronic obstructive pulmonary disease: is it a hidden predictor of survival and a risk factor for immune-related adverse events?". This article explores the complex and multifaceted interactions between the immune system and COPD and NSCLC. In patients with both diseases, COPD can impair the immune response against cancer cells by reducing or suppressing the activity of immune cells or altering their cytokine profiles. Anticancer treatments can also affect the immune system. and aggravate the symptoms of COPD by causing inflammation and fibrosis in the lungs. In this article, an attempt is made to understand the interplay between these two pathways and the efficacy of immunotherapy in NSCLC patients with COPD, but several points need clarification.
Comment:
1. The authors should include a description and discussion of the efficacy of regulatory T cells (Treg)/T helper 17 (Thl7) in non-small cell lung cancer (NSCLC) patients with chronic obstructive pulmonary disease (COPD)
2. Chronic obstructive pulmonary disease (COPD) and inflammatory events underlie non-small cell lung cancer (NSCLC), T helper 1 and T helper 2 cytokines, and type 1 and type 2 macrophages (M1 and M2) in NSCLC with and without COPD Differential expression is found in patients' lung tumors and blood, and the M1/M2 ratio specifically may affect their survival. Authors should include narratives and discussions that enhance it.
Comments on the Quality of English LanguageMinor editing of English language required
Author Response
We thank the Reviewer for her/his comments. A poit-by-point response is attached as a separate file.

Round 2
Reviewer 1 Report
Comments and Suggestions for Authors
Page 4, lines 136-137: “Among the inflammation-related transcription factors regulating PDL-1 overexpression stands interferon-γ (IFN-γ)…” IFN-γ is not a transcription factor.
Page 9, lines 369-373: “Indeed, in patients with patients with advanced NSCLC, some authors described a prevalence 69% of patients withof COPD (aboutranging from 39%), toemphysema (about 59%) or both in the NSCLC patients followed [86] while others reported an incidence of COPD in 50% patients with advanced NSCLC [119,120].”
Comments on the Quality of English LanguageMinor editing of English is required
Author Response
Dear Reviewer,
we are sorry for the oversight, of course IFN-g is not a transcription factor and emended the text accordingly (page 4, lines 136-138).
As regards page 6, the text should be read without track changes in "simple comments" modality. In fact, it only states "Indeed, in patients with advanced NSCLC, some authors described a prevalence of COPD ranging from 39% to 50% [119,120]. " The repetition of "with patients" was indeed present and has been removed.
Thanks for your careful check